# Clinical Effects of Faecal Microbiota Transplantation as Adjunctive Therapy in Dogs with Chronic Enteropathies—A Retrospective Case Series of 41 Dogs

**DOI:** 10.3390/vetsci10040271

**Published:** 2023-04-03

**Authors:** Linda Toresson, Thomas Spillmann, Rachel Pilla, Ulrika Ludvigsson, Josefin Hellgren, Gunilla Olmedal, Jan S. Suchodolski

**Affiliations:** 1Department of Equine and Small Animal Medicine, Faculty of Veterinary Medicine, Agnes Sjöberginkatu 2, Helsinki University, 00014 Helsinki, Finland; 2Evidensia Specialist Animal Hospital, Bergavagen 3, 25466 Helsingborg, Sweden; 3Gastrointestinal Laboratory, Department of Small Animal Clinical Sciences, Texas A&M School of Veterinary Medicine & Biomedical Sciences, 4474 TAMU, College Station, TX 77843, USA

**Keywords:** faecal microbiota transplantation, dog, chronic enteropathy, non-responsive enteropathy, dysbiosis, dysbiosis index

## Abstract

**Simple Summary:**

Chronic enteropathy is common in dogs, but not all dogs will respond satisfactorily to standard treatment. A few case reports suggest that faecal microbiota transplantation (FMT) can be used in dogs if standard treatments fail. Here, we report the effects of faecal microbiota transplantations in 41 dogs with chronic enteropathy, with a follow-up time of 3–41 months. None of the dogs had previously responded satisfactorily to standard treatment. Most dogs received three FMT treatments with 10–20 days’ interval in between. The treatment was given in parallel with the maintenance medical treatment and diet the dog was already being treated with. After treatment, 31/41 dogs had decreased clinical signs, with most dogs showing less diarrhoea and/or becoming more active. From 16 dogs, faecal samples were taken before the first FMT. Dogs with a more severe dysbiosis, as indicated by a severely increased faecal dysbiosis index, responded more poorly to FMT compared to those dogs with milder form of dysbiosis. Our study supports previous case reports that faecal microbiota transplantation can be useful as an adjunct to the standard treatment of dogs with chronic enteropathy.

**Abstract:**

Chronic enteropathies (CE) are common in dogs, but not all affected dogs respond to standard therapy. Successful responses to faecal microbial transplantation (FMT) in dogs with non-responsive CE have been reported in two case series. The objective of this retrospective study was to describe the clinical effects of FMT as an adjunctive therapy in a larger population of dogs with CE. Forty-one dogs aged 0.6–13.0 years (median 5.8) under treatment for CE at one referral animal hospital were included. Dogs were treated with 1–5 (median 3) FMTs as a rectal enema at a dose of 5–7 g/kg body weight. The canine inflammatory bowel disease activity index (CIBDAI) was compared at baseline versus after the last FMT. Stored faecal samples (*n* = 16) were analysed with the dysbiosis index. CIBDAI at baseline was 2–17 (median 6), which decreased to 1–9 (median 2; *p* < 0.0001) after FMT. Subsequently, 31/41 dogs responded to treatment, resulting in improved faecal quality and/or activity level in 24/41 and 24/41 dogs, respectively. The dysbiosis index at baseline was significantly lower for good responders versus poor responders (*p* = 0.043). Results suggest that FMT can be useful as an adjunctive therapy in dogs with poorly responsive CE.

## 1. Introduction

Chronic enteropathy is a prevalent disorder in dogs, and the majority of the affected dogs respond to evidence-based treatment protocols [1]. However, studies have shown that approximately 15–43% of dogs with CE have non-responsive enteropathy (NRE), with a poor long-term prognosis and a high risk of euthanasia [2,3]. A recent meta-analysis comparing the intestinal microbiome in healthy dogs vs. dogs with gastrointestinal (GI) disorders concluded that the majority of dogs with GI diseases have dysbiosis [4]. This dysbiosis is characterized by a significantly decreased diversity and reduced abundance of *Faecalibacterium*, *Fusobacterium*, *Blautia*, *Turicibacter* and *Clostridium hiranonis*, as well as an increased dysbiosis index and abundance of *E. coli*, compared to healthy dogs. A potential new treatment option in NRE is faecal microbiota transplantation (FMT), which in theory can directly address dysbiosis. Faecal microbial transplantation is used to transfer the intestinal microbiome from a healthy donor to a recipient with a disease, in order to improve the microbiome and decrease disease activity. In people with recurrent *Clostridioides difficile* infection, FMT is more effective than antibiotic treatment [5]. Furthermore, FMT has been used in two placebo-controlled studies on ulcerative colitis in people [6,7]. In the largest study, repeated rectal FMTs led to remission in 11/41 of the patients receiving FMT, compared to 3/40 patients receiving the placebo [6]. In a smaller study, repeated oral FMTs led to remission in 8/15 of the patients receiving FMT, compared to 3/20 receiving the placebo [7]. A Cochrane meta-analysis stated that “Faecal microbiota transplantation may increase the proportion of participants achieving clinical remission in ulcerative colitis” [8]. Faecal microbiota transplantation also appears promising in treating Crohn’s disease [9,10].

In dogs, FMT significantly reduced time to recovery and hospitalization time in puppies with parvovirosis in a randomized study [11]. Furthermore, one single FMT improved faecal scores to the same extent as 7 days of metronidazole in dogs with acute diarrhoea at day 7, and helped to restore the intestinal microbiome during the following 28 days [12]. Faecal microbiota transplantation did not, however, have any positive clinical effect in a small placebo-controlled study of eight dogs with acute haemorrhagic diarrhoea syndrome [13]. Currently, the effects of FMT on chronic diarrhoea in dogs are available in one prospective study and a few peer-reviewed case series or case reports [14,15,16,17,18]. In the prospective study, a clinical response to FMT was noted in 20/27 dogs with CE [18]. The Canine Chronic Enteropathy Clinical Activity Index (CCECAI) was significantly decreased 15 days after ending one month of treatment with daily oral freeze-dried FMT capsules, compared to baseline. The clinical outcome after day 15 was not reported. In the case series, nine dogs with NRE had a significant decrease in the canine inflammatory bowel disease index (CIBDAI) three days after one single FMT [16,19]. In a very recent worldwide observational study of small animal clinicians’ experiences with FMT, 33/115 responding clinicians had experience with FMT [20]. Chronic enteropathy was the most common indication to perform FMT. The clinical response to FMT for various indications was reported as “mixed” by 52% or “good” by 33% of the responders. However, dose, number of FMTs given and interval between administrations varied largely among responders, making it very difficult to assess the efficacy. A protocol for using lyophilized FMT capsules in CE is available [18]. Lyophilized capsules have, however, very limited availability in many countries. Data on FMT treatment as a rectal enema in a larger population of dogs with CE, using a standardized protocol, are still lacking. The main purpose of this study was to evaluate the clinical effects, based on CIBDAI, of FMT as adjunctive therapy in dogs with CE, and provide longitudinal data on outcome. The secondary goal was to compare the dysbiosis index (DI) at baseline and over time in dogs responding vs. not responding to FMT, in those animals where stored faecal samples were available [21].

## 2. Materials and Methods

### 2.1. Study Design and Ethics Approval

This study was a retrospective review of medical records at the Evidensia Specialist Animal Hospital in Helsingborg, Sweden (ESAHHS). The inclusion period lasted from March 2019 to July 2022. The procedure of FMT and principle behind it was explained to all dog owners prior to informed owner consent, and all but 5 dog owners were also provided with written information on FMT. Naturally voided faecal samples were stored after informed owner consent from samples brought to the ESAHHS for parasite screening prior to FMT, and/or from faecal samples brought to ESAHHS in connection with each visit for FMT. Naturally voided faecal samples from dogs can be collected without formal ethical approval in Sweden.

### 2.2. Animals

Privately owned dogs with CE not responding satisfactorily to elimination diet, probiotic and/or immunosuppressive treatment, available for follow-up for at least 3 months post FMT, were included. Dogs with CE had a history of persistent or intermittent clinical signs of GI disease for a minimum of 8 weeks. The previous work-up of these dogs included exclusion of extra-intestinal disease with ultrasonography, clinical chemistry, including basal cortisol, haematology, parasitology, and histopathologic evidence of GI mucosal inflammation. Dogs lacking intestinal biopsies but fulfilling the other criteria were also included if they had a history of GI signs for more than 12 months. Furthermore, dogs that were clinically stable but required maintenance doses of corticosteroids that were associated with marked side effects were also included, as well as dogs with frequent flare-ups in need of antibiotics to control the clinical signs. Exclusion criteria were starting a new immunosuppressive treatment, changing the diet, or increasing the dose of maintenance therapy in parallel with FMT, the detection of intestinal parasites during FMT treatment, or incomplete medical records.

The three donor dogs were clinically healthy, staff-owned dogs with a body condition score of 4–5/9. One was a neutered male Golden Retriever, which served as a donor from 7 to 10 years of age. This dog was used as a donor in over 90% of the transplantations. The other two donors were a four-year-old intact male mixed breed terrier-type dog and an intact female Labrador Retriever that started serving as a donor from one year of age. They were fed a premium maintenance dry food diet, and had never been treated with antibiotics, non-steroidal anti-inflammatory drugs or immunosuppressives. The donor dogs had a CIBDAI below 3, a DI below −3 [21] and were negative for extended beta-lactamase resistant *E. coli*. They were screened for faecal parasites and with the dysbiosis index once yearly. The main donor dog had a high abundance of the short-chain fatty acid (SCFA)-producing bacteria *Fusobacterium* and *Faecalibacterium*, measured by qPCR as part of the dysbiosis index on separate occasions (Figure 1). One of the diseased dogs included lived a long distance from ESAHHS and was, for practical reasons referred to the Anicura Albano Animal Hospital in Stockholm for the first 2 treatments, but was given a third FMT at the ESAHHS. The donor dog in Stockholm had a DI of −6.2 and was screened in the same manner as the donor dogs at ESAHHS.

### 2.3. FMT Procedure

Stools from the donor dogs were collected daily to every other day into plastic bags, labelled with date and weight, frozen and stored at −20 °C until processing for FMT within 48 h before each procedure. The maximum storage time for frozen faeces was 3 months. Food was withheld prior to the procedure for a minimum of 6 h, but the recipients were allowed to drink water. Just prior to the procedure, the recipient was walked for 30–60 min in order to defecate. A low dose of acepromazine (0.1 mg/kg) was administered subcutaneously 15 min prior to the procedure if no contraindications were present. Dogs with a known heart condition, or heart murmur for unknown reasons, were not premedicated.

The FMT protocol was based on a previously published protocol by Chaitman et al. (2020) [12]. Fresh frozen faeces at a dose of 5–7 g/kg body weight of the recipient were thawed in a designated fridge for 4–24 h prior to the procedure. The transplant was then stored at room temperature for 0.5–1 h, blended with a designated blender and mixed with 20–120 mL of sterile saline until a desirable texture (a consistency that could be passed through the syringe and rectal catheter with mild to moderate pressure) of the transplant was achieved. Thereafter, the transplant was filtered through a sieve. The filtered transplant was aspirated with 60 mL sterile syringe(s) and administered rectally using a 16 French phthalate-free PVC catheter. If the transplant was going to be used in the days following preparation, it was stored in filled and capped syringes in a designated fridge for a maximum of 72 h. After the FMT, the dog owners were instructed to minimize the dog’s physical exercise and withhold food for 4–6 h.

The recommended treatment plan was 3 FMTs with 10–20 days interval in between, but dog owners could decline further treatment at any time point.

### 2.4. CIBDAI, Definition of Clinical Response and Outcome

The canine IBD activity index was calculated at baseline and between 7 and 21 days after the 3rd FMT. For dogs only receiving 1 or 2 FMTs, the follow-up CIBDAI was calculated between 7 and 21 days after the single FMT or the last of the 2 FMTs. The questions which required the calculation of the CIBDAI were all part of the clinical history electronic template at ESAHHS. Dog owners graded the quality of the faeces by using a faecal scoring chart during clinical history taking [22]. The clinical response to FMT was graded according to Figure 2. Regarding outcome, dogs were followed from FMT 3 until the time of writing. For dogs only receiving 1 or 2 FMTs, the dogs were followed from the day of the single FMT or the last of the 2 FMTs. Flare-ups severe enough for the dog owners to contact ESAHHS and any increase in the maintenance dose of anti-inflammatory treatment were noted, and whether repeat FMTs were given. Furthermore, any change in the treatment protocol was noted, including changing the diet or medication.

### 2.5. Storage and Shipping of Faecal Samples

Faecal samples were stored in −20 °C at ESAHHS and shipped on dry ice to the Gastrointestinal Laboratory at the Texas A&M College of Veterinary Medicine and Biochemical Sciences with priority shipment every 6 to 12 months. The condition of the samples was recorded on arrival.

### 2.6. Bacterial qPCR Analysis

The samples were analysed with bacterial qPCR analysis as previously described [19]. Dysbiosis was evaluated using a previously published index, that has been shown to significantly differ between healthy dogs and dogs with chronic enteropathies. Faecal samples from dogs that were under treatment with antibiotics, or had been treated with antibiotics within 4 weeks, were excluded from qPCR analyses.

### 2.7. Statistical Analysis

All data analyses were performed using a commercially available statistical software package (GraphPad Prism 6.0, GraphPad Software). For normality testing, the D’Agostino and Pearson omnibus normality test was used. Neither CIBDAI pre and post FMT nor the comparison of CIBDAI at baseline in good responders versus short-lasting and poor responders combined showed normal distribution. For that reason, the former data were analysed with a Wilcoxon matched-pairs signed rank test, and the latter using the Mann–Whitney test. The comparison of DI at baseline in good responders versus short-lasting and poor responders combined followed a Gaussian distribution. Consequently, an unpaired *t*-test with equal standard distribution was used for statistical analysis. A *p*-value below 0.05 was considered statistically significant.

## 3. Results

### 3.1. Animals

Forty-one dogs aged 0.6 to 13 years of age (median 5.8) matched the inclusion criteria. The most common breed was German Shepherd (4/41), followed by Golden Retrievers (3/41) and mixed breed dogs (3/41) (Table 1). Twenty-eight dogs were male, of which 15 were intact and 13 castrated, and 13 dogs were female, of which 8 were spayed and 5 were entire female dogs. The most common reasons to perform FMT were diarrhoea (32/41), lethargy (18/41), difficulties tapering corticosteroids to tolerable maintenance doses (13/41) or various signs of abdominal pain and discomfort (13/41). The dogs had been treated for CE for 1–110 (median 20) months prior to the first FMT.

Intestinal biopsies were available from 34/41 dogs (Table 2). The biopsies were collected endoscopically in 33 dogs and via laparotomy in 1 dog and were graded according to the WSAVA guidelines [23]. The most common histopathologic diagnosis was mild to moderate lymphocytic-plasmacytic enteritis and/or colitis in 22/34 dogs (Table 2). Biopsies from the stomach was available in 32 dogs, from the small intestine in 34 dogs and from the large intestine in 27 dogs.

Regarding phenotype, most dogs (24/41) had partially immunosuppressive responsive enteropathy (P-IRE). These dogs responded to some degree to immunosuppressive treatment, but the treatment was not enough to control clinical signs to an acceptable level. Seven dogs had IRE, but required corticosteroid doses that were associated with unacceptable dermatological side effects (4/7) such as calcinosis cutis or very thin, frail cutis that lacerated frequently, lethargy and/or severe muscle atrophy (3/7). One of these dogs was also suffering from protein-losing enteropathy (PLE) with a serum albumin concentration below 20 g/L. Five dogs had non-responsive enteropathy (NRE) with a very poor response to various diets and treatment protocols. One of these dogs was also suffering from PLE. Three dogs had P-IRE and antibiotic responsive enteropathy (ARE), of which two of three dogs required frequent courses of antibiotics to control flare ups and one of three dogs required both corticosteroids q 24 h, chlorambucil q 48 h and metronidazole q 24 h to have some degree of control of the clinical signs. One dog had responded to a hydrolysed diet combined with a multi-strain probiotic, but had a severe flare up of gastrointestinal signs after treatment with antibiotics for a urinary tract infection. Lastly, a French Bulldog diagnosed with histiocytic ulcerative colitis (HUC) and a mixed lymphocytic-plasmacytic and eosinophilic enteritis was included. This dog had responded to 6 weeks of enrofloxacin, but relapsed with diarrhoea 21 days after antibiotics were completed. Besides the two dogs with PLE previously mentioned, none of the other dogs had a serum albumin concentration below 20 g/L.

Thirty-eight dogs were under treatment with corticosteroids at inclusion, and twenty-three of those dogs were treated with second line immunosuppressives in parallel with corticosteroids (Table 1). Most dogs (28/41) were fed a hydrolysed protein diet (Table 1).

### 3.2. FMT and Clinical Response

Treatment was in general easy to perform and well accepted by the dogs. One dog defecated a large volume within 30 min after FMT in the car on the way home, but all other dogs had an owner-reported minimum retention time of 1–15 h. Thirty dogs (71%) received three FMTs as a series of treatment, six dogs received two FMTs, two dogs received five FMTs, two dogs, both non-responders, received one FMT, one dog received four FMTs and one dog received more than five FMTs.

In thirty-one of forty-one dogs (76%), FMT was associated with an improvement in clinical signs. Twenty-six dogs had a good response, five dogs had a short-lasting response, and ten dogs were non-responders to FMT. Twenty-three dogs showed a further improvement of clinical signs after FMT two or three. All responders showed some degree of clinical improvement after FMT one and/or FMT two. The CIBDAI range (median) prior to treatment in all dogs was 2–17 (6), which decreased to 1–9 (2) after FMT three. For dogs only receiving one or two FMTs, the follow-up CIBDAI was calculated after the single FMT or the last of the two FMTs. The difference was statistically significant (*p* < 0.0001, Figure 3).

Disease activity at baseline was further compared between good responders (*n* = 26) versus short-lasting and poor responders combined (*n* = 15), but there was no significant difference between the groups (Figure 4). At baseline, CIBDAI range (median) was 3–15 (5) in the good responders and 2–17 (6) in poor responders (*p* = 0.55).

Regarding histopathology reports and response to FMT, two of four dogs with severe inflammation of the small or large intestine had a good response, one of four had a short-lasting response and one of four had a poor response.

Improved faecal quality and increased activity level were the most common improvements, noted in 24/41 dogs, respectively. Twenty-three of these dogs had diarrhoea prior to FMT. The remaining dog showed signs of pain upon defecation, with dry and firm faeces mixed with mucus, which after FMT resolved to normal stool and defecation without signs of pain. Improved activity level was expressed as taking more initiatives to play and interact with the dog owner or other dogs, being more active during walks, walking in front of the dog owner instead of behind, wanting to go for longer walks and spending less time sleeping during daytime. Of the 24 dogs being reported as more active, 16 had been reported as lethargic by the dog owners prior to FMT. The remaining eight dogs had been described as having a normal activity level prior to FMT, but the dog owners all remarked that the dogs were markedly more active on an open question on how the dog was doing after FMT.

Twelve of forty-one dogs could decrease the dose of corticosteroids or stop antibiotics after FMT. In ten out of twelve dogs, maintenance doses of corticosteroids could be tapered to doses that had not been possible prior to FMT. The remaining two dogs had frequent flare-ups, during which only metronidazole or tylosin could stop the diarrhoea. Both dogs could stop the frequent use of antibiotics after FMT for 3–20 months. Lastly, nine of forty-one dogs experienced less frequent flare-ups or milder flare-ups after FMT, eight dogs previously underweight gained weight after FMT and six dogs with hyporexia at baseline showed increased appetite post FMT.

### 3.3. Adverse Effects

Ten dogs, of which seven were responders and three were non-responders, experienced side effects after FMT. Seven dogs had diarrhoea or worsening of diarrhoea within 48 h after FMT, which normalized within 2–3 days without intervention in all but one non-responder. Four of these dogs only reacted to one of three FMTs. Of the three out of ten remaining dogs, all responders to FMT, one had a flare-up of diarrhoea and occasional vomiting 1 week after three of three FMTs, which lasted up to 48 h. The dog with HUC had 2–3 days of marked flatulence, stinking faeces, and mild vomiting after FMTs one and two, but not after FMT three. One dog experienced intense rectal straining and discomfort starting 2 h after the first FMT and lasting for 4 h. This dog was premedicated with a higher dose of acepromazine and a rectal suppository (Xyloproct, Aspen Nordic, Ballerup, Denmark) containing hydrocortisone and lidocaine 10 min prior to FMTs two and three. Additionally, the faecal transplant was mixed with a smaller amount of saline to decrease the volume. No adverse effects were noted after FMT two and three in this dog.

### 3.4. Bacterial qPCR Results

All faecal samples were in good condition and still frozen when arriving at the Gastrointestinal Laboratory. Sixteen baseline faecal samples, collected within 1 week prior to the first FMT, matched the inclusion criteria. Eight baseline samples were from good responders, and eight were from poor responders. The dysbiosis index was significantly lower (range (median)) at baseline in good responders (−5 to 3.3 (0.4) compared to poor responders (−0.7 to 6.7 (2.8), *p* = 0.043, Figure 5). 

### 3.5. Long-Term Outcome

A follow-up period of 3–40 (median 17.5) months was available from all dogs. Selected data on long-term follow-up are available in Table 3. More detailed information is available in the following sections.

#### 3.5.1. Good Responders

Sixteen of the twenty-six good responders had one to nine (median three) additional FMTs during a time period of 6–40 (median 25) months after the first series of treatment. Reasons for repeated treatment were recurrence of clinical signs in ten of sixteen dogs, adding a booster dose 3–4 months after the first series to potentially prolong the effect in five of sixteen dogs, and trying to treat newly emerged clinical signs in one dog. A positive effect of additional FMTs on clinical signs were noted in 15/16 dogs, although two dogs did not show the same degree of response as after the initial FMTs according to the dog owners.

In nine good responders, it was possible to decrease the maintenance dose of corticosteroids after FMT, and continue with decreased doses during the entire follow-up period of 6–39 (median 19) months. Some dogs had minor flare-ups during that time, during which the maintenance dose of corticosteroids had to be increased occasionally and temporarily. None of the dogs that experienced flare-ups required doses of corticosteroids as high as those prior to FMT to control clinical signs. At the time of writing, the dog with the longest follow-up period in this group has been markedly more clinically stable than prior to FMT for 39 months. It has not been treated with any additional FMT and receives a maintenance dose of corticosteroids that is reduced by 68% compared to dosing prior to FMT.

Two of the dogs requiring frequent use of antibiotics prior to FMT and the dog with HUC were good responders. One of the dogs that required frequent courses of tylosin prior to FMT was stable for 20 months without antibiotics after FMT. After 20 months, a more severe flare-up occurred that could not be controlled by increasing the maintenance dose of budesonide. Tylosin was then used for 7 days, and the dog was thereafter successfully treated with a new series of three FMTs. At the time of writing, 10 months after the second series of FMT, the dog has not needed any more antibiotics. In another dog that was treated with metronidazole every other day prior to FMT, it was possible to stop the antibiotics for 3 months post FMT. The dog was then started on gabapentin for non-gastrointestinal reasons, which was associated with a severe flare-up of diarrhoea within 48 h. At that time point, the dog’s owners started to use metronidazole again. Lastly, the dog with HUC and a recurrence of colitis 21 days after ending enrofloxacin responded to FMT and was free of clinical signs without any other drugs during the whole follow-up period of 24 months.

Five of the good responders were euthanized during the follow-up period at the age of 6.0–13.3 (median 10.3) years. All dogs were euthanized for non-gastrointestinal disorders, such as osteoarthritis (2/5), laryngeal paralysis and repeated aspiration pneumonias (1/5), biting the dog owner (1/5) and vestibular disease (1/5).

#### 3.5.2. Short-Lasting Responders

Of the five dogs with a short-lasting response to FMT, one was treated with monthly FMTs to keep the clinical signs somewhat controlled. This dog had a CIBDAI of 17 prior to FMT and 9 after FMT. After 9 months, the dog was started on one lyophilized FMT capsule q 24 h orally from the same donor dog, to increase the time interval between the rectal enemas. This had some positive effects on clinical signs, but it was only possible to extend the interval between rectal enemas by 7 days. The dog was euthanized 9 months later due to severe dyspnoea. On post mortem examination, marked pulmonary fibrosis and calcifications were found. The calcifications could have been caused by daily treatment with corticosteroids during several years, since tapering was associated with worsening of diarrhoea. One dog was treated with repeated FMTs and then switched to commercially available oral FMT capsules (DoggyBiome gut restore supplement; AnimalBiome, Oakland, CA, USA) daily and intermittent booster doses with rectal enema every fourth to fifth month. This was associated with a reduced number of flare-ups compared to prior to FMT, and the dog is generally stable at the time of writing, 15 months after the first FMT. The third dog had two more FMTs and responded to both treatments, but the effect only lasted 4 weeks each time. Repeated faecal dysbiosis indexing showed that the dog had persistently low amounts of *Clostridium hiranonis*, a bacteria responsible for the conversion of primary bile acids to secondary bile acids in dogs [24]. Persistent low abundance of *Clostridium hiranonis* has been associated with bile acid dysmetabolism in dogs with chronic enteropathy in a recent study [25]. For that reason, treatment with bile acid sequestrants was started. This treatment had a very good clinical effect, and the dog was stable for 5 months. Then, a per acute incident of severe abdominal pain, diarrhoea, inability to move and poor response to stimuli occurred, and the dog owner opted for euthanasia (without necropsy). The fourth dog initially had a very good response to FMT and could be managed with a dose of corticosteroids lower than before for 3 months. Then, three severe flare-ups of diarrhoea and melena occurred 3 to 9 months after FMT, which led to euthanasia. The fifth dog had a good response for 2 months post FMT, which was followed by an exacerbation of clinical signs. The dog was euthanized 3.5 months after the last FMT.

#### 3.5.3. Non-Responders

In dogs not responsive to FMT, four out of ten were euthanized within 4.5–25 months due to non-responsive disease. Three of these four dogs had been treated with bile acid sequestrants to exclude bile acid diarrhoea (BAD) for 4–8 weeks, without any clinical effect. One additional non-responder did not respond to cholestyramine either, but responded to switching the corticosteroids from methylprednisolone to budesonide. Two additional non-responders had an excellent response to bile acid sequestrants and were presumed to have BAD [26]. For the remaining three non-responders, one responded to switching the single protein source of the balanced home-cooked diet from horse to rabbit, one slowly became more clinically stable on the baseline treatment over 24 months, and the remaining dog was lost to follow-up 3 months after FMT.

## 4. Discussion

In this retrospective study, we report the clinical effects of FMT as an adjunctive therapy in 41 dogs with poorly responsive CE. The canine IBD activity index was significantly reduced after repeated FMTs. A positive clinical response was noted in 31/41 of the dogs after FMT, of which 26/41 had a good response, 5/41 had a short-lasting response and 10/41 no response. To the best of our knowledge, this is the largest study on the clinical effects of FMT in dogs with CE to date.

Improved faecal quality was one of the expected outcomes, based on previous work in dogs and humans with chronic diarrhoea and gastrointestinal inflammation [6,8,9,14,15,16,17]. The increased activity level in 24/41 dogs was slightly more surprising, especially in 8/24 dogs that were not described by the dog owners as lethargic or less active prior to FMT. Potential explanations for this finding could be reduced inflammation, associated with decreased fatigue and chronic pain, tapered doses of corticosteroids or other drugs, a modulation of the intestinal metabolome, alterations of the gut–brain axis, or even a placebo effect driven by the dog owner’s strong desire for a positive outcome. Fatigue and depression are common consequences of inflammatory bowel disease or irritable bowel syndrome (IBS) in people [27,28]. Interestingly, in studies on the effects of FMT in people with IBS and IBD, self-assessed quality of life improved after FMT for 3–6 months. A similar effect was seen on fatigue post FMT, with improvement lasting up to 6 months [29,30,31]. Several intestinal microbes can synthesize neurotransmitters, such as γ-amino butyric acid (GABA), glutamate, noradrenaline and dopamine [32]. This can have a local effect in the gut, but also modulate the brain and behaviour. Furthermore, the beneficial microbes of the intestinal microbiota produce SCFAs, which stimulate tryptophan hydroxylase 1 in the enterochromaffin cells, resulting in 5-HT (serotonin) synthesis and secretion. Serotonin is one of the most important modulators of the gut–brain axis, and 95% of 5-HT is located in the gut [33,34]. Lethargy is a common side effect of corticosteroid treatment in dogs [35]. Decreasing the maintenance dose of corticosteroids could be associated with increased activity level. However, the corticosteroid dose was typically not tapered until after the last FMT, at which time point the dogs had already shown more active and playful behaviour. Lastly, a placebo effect in some dogs could not be excluded. The evaluation of the CIBDAI was based on owner-reported data, which makes this assessment tool vulnerable to owners’ subjective perceptions. Several dog owners did, however, report that it was not just the closest family that had noticed this change, but also distant relatives, dog walkers and staff at dog day care centres unaware of recent treatment with FMT. This makes it less likely that all 24/41 dogs reported as being more active after FMT could be solely attributable to a placebo effect. Furthermore, the dogs included in this study were not dogs newly diagnosed with CE, which could hypothetically progress to a self-resolution, but dogs that had already received multiple treatments without success. Consequently, self-resolution or placebo response would be exceptions and not to be expected in 31/41 dogs.

The antibiotic-sparing effect of FMT in three out of four dogs where FMT was used mainly or partly to avoid repeated use or reduced use of antibiotics is encouraging. Antibiotics, especially broad-spectrum antibiotics such as tylosin and metronidazole, cause dysbiosis that can last for several weeks or up to months in dogs [36,37,38]. In people, the use of antibiotics in early life was shown to increase the risk of later being diagnosed with IBD in a large, recent prospective case–control study in 194,163 individuals [39]. The odds ratio increased with broad-spectrum antibiotics and multiple antibiotic dispensations. The sharing of antimicrobial resistance genes has further been shown to occur between dogs and humans living in the same household [40,41,42]. Although not used in people, tylosin induces macrolide resistance, and macrolides are critically important antimicrobials in human health [43,44]. The good response to FMT in the dog with relapsing HUC is extra promising. Dogs with relapsing HUC have a guarded prognosis, and are often treated with antibiotics that are prohibited for use in animals within the European Union [45,46]. Two years after FMT, this dog has still not had any relapse of persistent diarrhoea.

The rationale for performing sequential FMTs was based on results from people with CDI and/or IBD, a master’s thesis on FMT in dogs with IBD and a conference abstract with follow-up data on dogs with chronic diarrhoea treated with a single FMT [6,47,48,49,50,51]. In people, several authors have suggested that repeated FMTs can increase the efficacy rates of FMT [7,47,48,49]. In the master thesis, microbiota analysis was performed every week for two months after one single FMT in dogs with IBD [50]. The dysbiosis index decreased drastically after FMT, but started to increase again after 3–4 weeks. Similar (unpublished) results were shown by Chaitman and co-workers [51]. In the abstract, a significant decrease of DI was seen one week after FMT in dogs with chronic diarrhoea. Four weeks after FMT, DI started increasing again in about a fourth of the dogs. Clinical experience from the current study also supports the use of multiple FMT. In 23/31 responders, further clinical improvement was noted after the second FMT compared to the first. In three of the good responders, a relapse of clinical signs was noted just prior to FMT two but was reversed after the second FMT. Lastly, the positive effect of repeated FMT in dogs experiencing a relapse of clinical signs at a later time point, after the initial sequential FMTs, has been in shown in people too [47].

Although only 16 samples were available for comparison between good and poor responders, our results suggest that dogs with a higher DI, indicating a more severe shift in the microbiome, may be less likely to respond favourably to FMT. A high DI is negatively correlated with microbial diversity [38]. Similar results have previously been shown in people with ulcerative colitis, in a placebo-controlled, double-blind study using rectal FMT or placebo enemas [6,52]. All patients were treated with 40 rectal enemas per person, either as FMT or placebo enema. Patients responding to FMT had a more diverse microbiome before and after FMT. The correlation between the intestinal microbiome of the recipient at baseline and the response to FMT was further described in studies of people with Crohn’s disease and recurrent *Clostridioides difficile* infection (CDI) [53,54]. Successful donor microbiota engraftment is likely easier in a less hostile gut environment than that of dogs and people with a pronounced shift of the intestinal microbiome, and severe depletion of beneficial microbes and corresponding metabolome. Another potential reason why responders to FMT had a low DI at baseline in this study may be related to concurrent medical treatment and diet. Most of the dogs were already under treatment with corticosteroids and had been fed a hydrolysed diet for an extended period. It has previously been shown that hydrolysed diets and corticosteroid treatment can reduce dysbiosis and induce a shift in the intestinal microbiome towards that of healthy dogs [55,56]. However, the dogs in those studies had a more favourable response to dietary or medical interventions than those seen in dogs included in this study.

A positive outcome following FMT in people has further been associated with an increase in beneficial metabolic pathways, such as SCFA biosynthesis and secondary bile acids post FMT [52,54]. Short-chain fatty acids are produced by several bacterial taxa during the fermentation of soluble dietary fibre. These metabolites have multiple beneficial effects on gut health, such as reducing inflammation by stimulating regulatory T cells, inhibiting the excessive signalling of Toll-like receptors, acting as an energy source for colonocytes, regulating intestinal motility, and maintaining an intact intestinal barrier via various mechanisms [33,57,58]. A lack of beneficial, SCFA-producing intestinal microbes can thus be associated with a pro-inflammatory state and loss of functionality. Although the analysis of faecal bile acids or other aspects of the intestinal metabolome were not included in this study, it is suspected that some of the poor responders or short-lasting responders had bile acid dysmetabolism and BAD. A marked positive clinical response to the bile acid sequestrant cholestyramine was seen in two non-responders [26] and one short-lasting responder. A previous study in dogs with corticosteroid-responsive chronic enteropathy showed that these dogs had a decreased abundance of *Clostridium hiranonis* and a decreased percentage of secondary bile acids at baseline compared to healthy dogs [25]. These changes normalized over two to three months with corticosteroid treatment in most, but not all dogs. As secondary bile acids inhibit the growth of *C. difficile*, *C. perfringens*, *E. coli* and other gut microbes [59], a lack of secondary bile acids may have contributed to a high DI in some non-responders. Similar bile acid dysmetabolism has been shown in people with inflammatory bowel disease, especially during flare-ups [60].

Chronic enteropathy is a multifactorial disease associated with multiple pathophysiological changes [61]. The mucus layer of the intestinal tract forms an important physical barrier against invading bacteria and toxic substances, and serves as a nutrient for some beneficial microbes [62]. This mucus layer becomes thin and degraded in people with IBD, which affects gut homeostasis and can contribute to dysbiosis [62,63,64]. Altered mucin gene expression and a decreased percentage of goblet cells has further been shown in miniature dachshunds with inflammatory colorectal polyps [65]. Examples of other pathophysiological changes described in CE are the atrophy of intestinal villous and microvillous, crypt lesions, lacteal dilation, impaired intestinal receptors and a lack of transport proteins [61]. All of these changes can result in malabsorption. Malabsorption is associated with an increased amount of macronutrients in the intestinal lumen, which can cause dysbiosis and diarrhoea, directly or indirectly [66,67,68]. Some of these parameters may also have contributed to the difference in DI between responders and non-responders, even if CIBDAI did not differ between good responders versus short-lasting and poor responders combined.

The microbial composition of the donor stool has also been associated with outcome in ulcerative colitis and CDI [52,54,69]. We speculate that the high abundance of beneficial microbes in the stool of our most frequently used donor dog was associated with the good outcome in the majority of dogs. The FMT dose used in this study was higher than in many studies in people, which also may have affected the outcome. However, in a recent review on FMT in Crohn’s disease, the dose of faeces used (<50 g or >50 g) did not appear to affect the efficacy of treatment [10]. Furthermore, in people, the route of administration (oral versus rectal) does not appear to be associated with the outcome of FMT for the indications available to date (CDI, ulcerative colitis and Crohn’s disease) [5,6,7,9,10].

The procedure used for FMT in the current study can easily be used in primary-care settings. Using fresh frozen faeces is convenient compared to fresh faeces, which may not be readily available. Frozen faeces were considered as effective as fresh in treating recurring CDI in a recent meta-analyses [70]. Freezing may also have a protective effect against potential faecal endoparasite eggs or oocysts of protozoan parasites that the donor dog could have contracted after the last screening [71,72,73]. No cryopreservation was used in this study, which may have negatively affected the microbial viability of the transplant. However, the clinical effect was still good in most dogs.

In general, the dog owners’ perception of FMT was very positive. Many of them expressed that they would rather come back for booster FMTs a few times per year than risk the dog reverting to the clinical status prior to FMT or having to add a potent second line immunosuppressive or intermittent courses of antibiotics. For that reason, booster doses were given to some dogs even if it was not obvious that it was clinically necessary. One of the dogs still alive today would have been euthanized in March 2019 if the dog had not responded to FMT, as the dog was refractory to standard evidence-based treatment. This dog had an excellent response to FMT.

Side effects were mild and self-limiting and were seen in both responders and non-responders. Consequently, short-lasting mild diarrhoea or other GI signs after FMT do not exclude a beneficial effect, and mild side effects do not appear to differ responders from non-responders in this study.

This retrospective study has several limitations. Most dogs were treated with a transplant that was processed within 6 h, but some dogs were treated with a transplant that had been processed up to 72 h previously. This may have affected the response to FMT. Information on the time span between processing the transplant and FMT was not available for individual dogs. Faecal samples at baseline were only available from 16/41 dogs, and faecal parameters were followed longitudinally only in a few dogs. The clinical follow-up period varied due to the retrospective nature of the study. There was no placebo group and dogs’ owners were not blinded to treatment, which could be associated with some level of placebo effect. However, some objective measurements were used, such as medication use, number of flare ups, and DI when samples were available. Many questions remain to be answered on the use of FMT for dogs with CE. More data are needed on how long clinical improvement usually lasts, although clinical experience from this study suggests that the response to treatment is very individual. Further research questions for future studies include how many sequential treatments should be given, if diagnostic tools can differentiate poor responders from good responders at baseline, and how the microbiome and metabolome are affected over time. Several of these questions will be addressed in an ongoing prospective study.

## 5. Conclusions

Faecal microbiota transplantation appears to be a valuable adjunctive treatment in dogs with CE. Clinical disease activity was significantly reduced after sequential FMT. Several dogs were successfully managed with reduced maintenance doses of corticosteroids after FMT. Faecal microbiota transplantation also had an antibiotic-sparing effect in some dogs. Treatment was repeated in several dogs after later flare-ups, but this was again associated with a good clinical response. Side effects were few, mild and transient.

## Figures and Tables

**Figure 1 vetsci-10-00271-f001:**
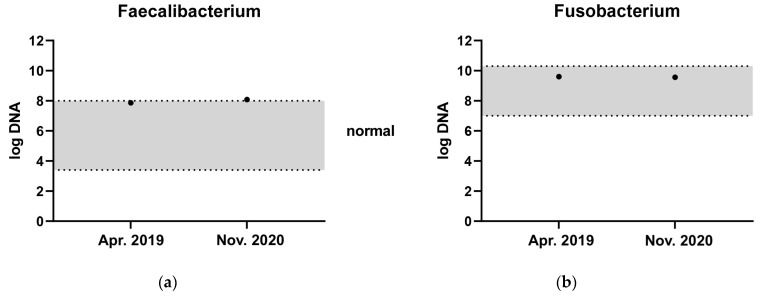
Faecal abundance of (**a**) *Faecalibacterium* and (**b**) *Fusobacterium* at two different time points from the most frequently used donor dog. The grey zone represents the reference interval.

**Figure 2 vetsci-10-00271-f002:**
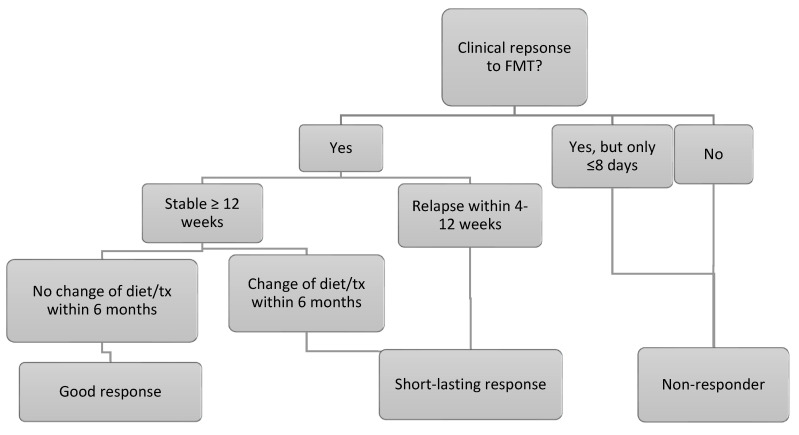
Flow-chart of the classification of response to FMT. Good responders also had to have a minimum CIBDAI improvement of 2 compared to baseline. If baseline CIBDAI was 4–5, indicating mild clinical disease activity, the response was graded as good if the CIBDAI was consistent, but maintenance therapy could be tapered to doses that had not been possible before.

**Figure 3 vetsci-10-00271-f003:**
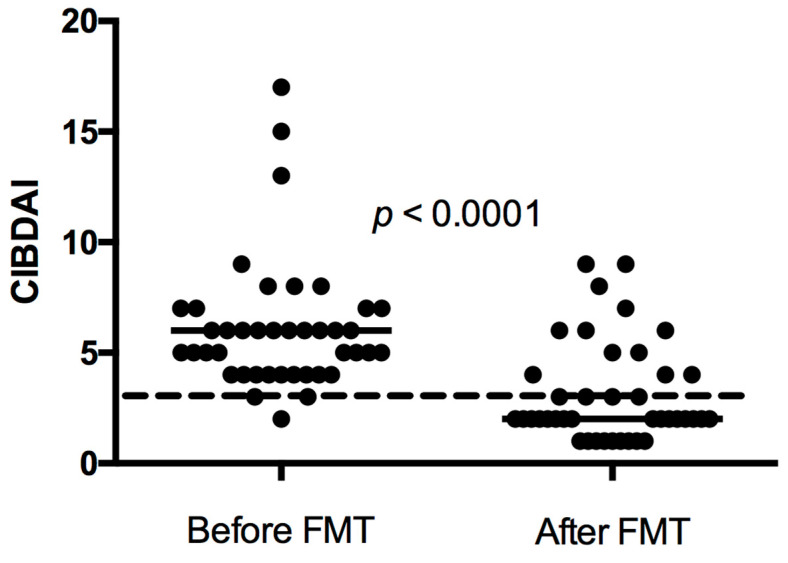
Canine inflammatory bowel disease activity index (CIBDAI) in 41 dogs with chronic enteropathy treated with faecal microbiota transplantation (FMT) before FMT and after FMT 3 in dogs receiving 3 or more FMTs. For dogs only receiving 1 or 2 FMTs, the follow-up CIBDAI was calculated after the single FMT or the last of the 2 FMTs. Short horizontal lines represent median. The long striped line represents CIBDAI of 3, which is the upper limit for clinically insignificant disease.

**Figure 4 vetsci-10-00271-f004:**
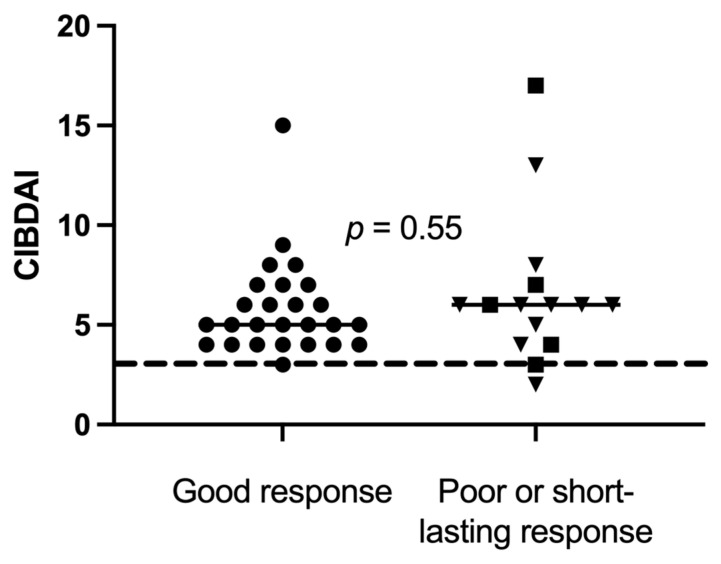
Canine inflammatory bowel disease activity index (CIBDAI) at baseline in 41 dogs, of which 26 good responders (●), 5 short-lasting responders (■) and 10 poor responders (▼), with chronic enteropathy, were stratified after clinical response to sequential FMT. Short horizontal lines represent median. The long striped line represents CIBDAI of 3, which is the upper limit for clinically insignificant disease.

**Figure 5 vetsci-10-00271-f005:**
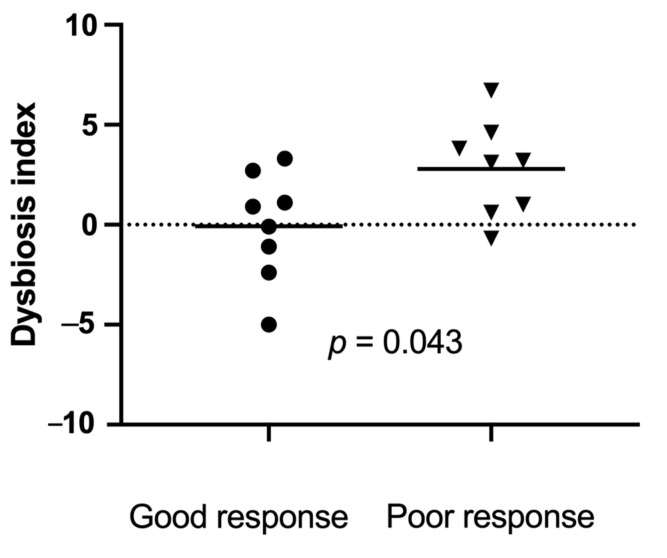
Dysbiosis index just prior to the first faecal microbiota transplantation (FMT) in 16 dogs with a good response ((●) *n* = 8) versus a poor response ((▼) *n* = 8) to FMT. Short horizontal lines represent mean. The dotted line at 0 represents the upper limit of the reference interval. Results above 2 represent a significant dysbiosis, and those between 0 and 2 represent mild dysbiosis.

**Table 1 vetsci-10-00271-t001:** Selected data at inclusion in 41 dogs with CE treated with FMT.

Parameter	Result (Range (Median) or Number)
Age (years)	0.6–13.0 (5.8)
Breeds ^a^	GSD 4, GR 3, MB 3, LR 2, RT 2, WH 2, STDPDL 2, ESS 2, MISC 21
Treated for CE (months)	1–110 (20)
Immunosuppressive treatment	CS ^b^ 15, CS + mycophenolate 8, CS + cyclosporine 7, CS + chlorambucil 6, CS + azathioprine 2
Miscellaneous treatment ^c^	Probiotics ^c^ 18, olsalazine 13, cobalamin 11, prebiotics ^d^ 11, cisapride 3, sucralfate 3, metronidazole 2, mirtazapine 2
Diet ^e^	Hydrolysed protein 28, single protein 11, GI ^f^ 2
Main reasons for FMT	Diarrhoea 32, lethargy 18, unable taper CS 13, abdominal pain 13, hyporexia 9, underweight 6, reduce use/need of antibiotics 3

^a^ GSD: German Shepherd Dog; GR: Golden Retriever; MB: Mixed breed; LR: Labrador Retriever; RT: Rottweiler; WH: Whippet; STDPDL: Standard Poodle; ESS: English Springer Spaniel; MISC: Miscellaneous breeds represented by 1 individual/breed ^b^ CS: corticosteroids (budesonide, methylprednisolone or prednisolone); ^c^ multistrain high dose probiotic (Sivomixx^TM^ or Vivomixx^TM)^; ^d^ multi-fibre blend or psyllium; ^e^ 38 dogs were fed premium kibble diets from major pet food companies, 3 dogs were fed a home-cooked balanced diet (horse and potato or moose and rice); ^f^ GI: gastro-intestinal (easily digestible diet).

**Table 2 vetsci-10-00271-t002:** Histopathology reports from gastrointestinal biopsies of 34 dogs with CE.

Location	Histopathological Findings	Severity	*n*
Stomach	LP ^a^ gastritis	Mild	12
		Moderate	5
		Severe	2
	Eosinophilic gastritis	Mild	3
	LP and partially eosinophilic gastritis	Mild	2
		Moderate	2
	Mixed cell type (eosinophils/LP/neutrophils)	Mild	1
		Moderate	1
	Fibrosis without inflammation	Mild	1
	Normal stomach	N/A ^b^	3
Small intestine	LP enteritis	Mild	9
		Moderate	11
	LP enteritis with multiple erosions	Severe	2
	LP and partially eosinophilic enteritis	Moderate	4
	Eosinophilic enteritis	Mild	3
		Moderate	2
	Normal small intestine	N/A	3
Large intestine	LP colitis	Mild	15
		Moderate	1
	LP colitis w multiple erosions	Mild	1
	LP and partially eosinophilic colitis	Mild	1
		Moderate	1
	LP and partially eosinophilic colitis w. erosions	Moderate	1
	Eosinophilic colitis	Mild	1
	Erosive/ulcerative colitis	Moderate	1
		Severe	1
	Histiocytic ulcerative colitis	Severe	1
	Normal large intestine	N/A	3

^a^ LP: lymphocytic-plasmacytic; ^b^ N/A: not applicable.

**Table 3 vetsci-10-00271-t003:** Long-term outcome (3–40 months) in 41 CE dogs treated with FMT as an adjunctive therapy.

Parameter (Number of Dogs)	Good Responders *n* = 26	Short-Lasting Responders *n* = 5	Non-Responders *n* = 10
Euthanasia for refractory GI ^a^ dx ^b^	0	3	4
Additional FMTs performed in	16	1	0
FMT due to recurrence of clinical signs	10	15	N/A
Adding a booster dose	5	3	N/A
Treating new clinical GI signs	1	0	N/A
Clinically stable > 4 w. ^c^ after additional FMT	15	0	N/A
Tapering of maintenance corticosteroid tx ^d^	9	1	0
Clinically long-term stable on decreased doses of corticosteroids	9	0	N/A
Stopping or reducing antibiotics	3/3	0/1	N/A
Treatment with cholestyramine	0	1	5
Response to cholestyramine	N/A	1	2

^a^ GI: gastrointestinal, ^b^ dx: disease, ^c^ w.: weeks, ^d^ tx: treatment.

## Data Availability

Data available on request due to restrictions, e.g., privacy or ethical reasons. As the study is retrospective, no owner consent has been signed, and the data are therefore not publicly available.

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
