# Peer review of "Clinical Effects of Faecal Microbiota Transplantation as Adjunctive Therapy in Dogs with Chronic Enteropathies—A Retrospective Case Series of 41 Dogs"

_vetsci, 2023, doi:10.3390/vetsci10040271_

Round 1

Reviewer 1 Report

This is a timely retrospective study of faecal microbiota transplantation (FMT) and its applicability to intestinal disorders in a well-powered study of 41 dogs. The title is appropriate and the abstract is a good summary of the study. Some minor grammatical improvements are required. The Introduction is well-structured, representing a solid summary of inflammatory intestinal conditions in dogs and the potential for improvement via FMT. 

Methods are appropriate and well described. The resolution of figure 1 could be improved. It is also not clear what the symbols on figure 1 represent. Results are described well. Tables and figures are clear and appropriate. The Discussion is extremely thorough, (possibly too long) but nevertheless provides some interesting comparisons with FMT in humans. 

Author Response

Dear reviewer,

We would like to thank you for your insightful, positive and very fast review of our manuscript. 

We have edited figure 1 regarding resolution and symbols, and made some edits to the text (see new version of the manuscript). 

Sincerely,

Linda Toresson on behalf of all authors

Author Response

Dear Reviewer,

We would like to thank you for your insightful and very fast review of our manuscript. We have attempted to respond to all suggestions and concerns to the best of our abilities and believe that the revised manuscript has gained clarity and improved quality. A special thanks for noticing the incorrect calculations! Please let us know if you have any further questions.

Sincerely,

Linda Toresson on behalf of all authors

General comments:

  1. We have changed the title and included ”case series”.
  2. We have added a scheme as suggested by the reviewer, to make the retrospective grouping easier to follow (see comment 10)
  3. Regarding the statement on dysbiosis index, please see comment 24 where this specific question is addressed.
  4. LINES 53-54 and LINES 70-71: “However, only case reports with a small number of CE dogs treated with FMT are available at present” and “Currently, the effects of FMT on chronic diarrhoea in dogs is only available in a few peer-reviewed case series or case reports”. In a recent peer-reviewed open-access article (DOI:10.3390/vetsci9090502), Innocente, Patuzzi et al., reported clinical efficacy of FMT in 27 CE-affected dogs. Clinical signs of improvement were observed in 74% (20/27) of CE-affected dogs, together with a statistically significant decrease in CCECAI (median value from 5 to 2 median)

Response: The sentence in lines 53-54 has been removed and the reference added

  1. LINES 60-62: Furthermore, in a placebo-controlled study in ulcerative colitis in people, repeated FMTs led to remission in 11/41 of the patients receiving FMT, compared to 3/40 patients receiving placebo 10. A Cochrane meta-analysis stated that ”Faecal microbiota transplantation may increase the proportion of participants achieving clinical remission in ulcerative colitis” 11. Recent literature on FMT human trial is missing, new findings show a higher efficacy e.g. DOI:https://doi.org/10.1016/S2468-1253(21)00400-3.

Response: Thank you, this reference has been added

  1. LINE 79-80: “Data on FMT treatment in a larger population of dogs with CE, using a standardized protocol, is still lacking.” As previously stated, Innocente, Patuzzi et al., 2022, (DOI:10.3390/vetsci9090502), reported the clinical efficacy of standardized protocol, in which FMT was administered daily for a month in freeze-dried form, with a dosage of 100 mg daily for dog body weight ≤ 10 kg, and 200 mg daily, for dog body weight > 10 kg.

Response: Thanks you, we have changed the wording to include this paper.

  1. LINE 112: Donors age and breed is not specified.

Response: This information has been added (line 166-168)

  1. LINE 116: “The donor dogs had a CIBDAI below 3, a DI below -3 and were negative for ex- 115 tended beta-lactamase resistant E.coli [19]”. Reference is related to DI, not to a protocol to test the presence of beta-lactamase resistant E.coli.

Response: We have moved the citation

  1. LINE 129-152: “The maximum storage time for frozen faeces was 3 months”,” Fresh frozen faeces at a dose of 5-7 grams/kg body weight of the recipient was thawed in a designated fridge for 4-24 hours prior to the procedure.” “If the transplant was going to be used during the following days after preparation, it was stored in filled and capped syringes in a designated fridge for a maximum of 72 hours.” The storage time of raw material and of intermediate products varies among cases; this could affect microbiota composition and FMT efficacy. If data on storage time are available, they should be correlated to the FMT efficacy.

Response: This information is unfortunately not available. We have added that as a limitation of the study

  1. LINES 154-175: The criteria chosen to define response are arbitrary and confusing, combining clinical signs (CIBDAI) with changes in diet or treatments, evaluating them sometimes in relation to the first FMT, sometimes to the last FMT.
    Please make a scheme to let the reader understand.

Response: The reason why we have included both CIBDAI and changes of treatment or diet after FMT is that CIBDAI only reflects disease activity at one timepoint, and this can be very transient.  Changes of diet and/or medication only occured in dogs that had a poor or short-lasting effect of FMT. If only CIBDAI would be used, more dogs would seem to have a good response. However, we believe that the longitudinal data is very important and is lacking in the current studies or case series/reports of FMT in dogs with CE. A scheme has been added as suggested

  1. LINES 165-175: Responders include two different groups used to create figures 3 and 4. Please verify that if the two groups are marked separately (different shapes or colors), they appear mixed, otherwise if they appear clearly separated, a reanalysis of the data should be considered.

Response: Different shapes of short-lasting and poor responders has been added to figure 3. The results were not clearly separated. Figure 4 now only consists of DI from good and poor responders. The DI of the single dog with a short-lasting response to FMT has been removed.

  1. LINE 171 Don’t understand the sentence.

Response: The sentence has been edited

  1. LINE 160: It is not clear if the POST-FMT CIBDAI was evaluated after the last FMT administration. If the evaluation time varies from case to case, a supplementary figure with an experimental timeline for each dog is required.

Response: We have clarified that CIBDAI after FMT was calculated between 7 and 21 days after the last FMT. We did call all the dog owners after the last FMT of the initial 3 FMTs (or, for those receiving only 1 or 2 FMTs, after the last (or only ) FMT). For several dogs, CIBDAI was the same as when they came for FMT 3, but some had further improvement after FMT 3. Since we have follow-up data for a minimum of 3 months from each dog after the last FMT, we do no believe that it is as crucial to know exactly which day between day 7 and 21 CIBDAI was calculated for individual dogs – only in the non-responders could a few of them experience a very transient improvement after FMT. The short-lasting responders all had a clinical improvement that lasted for at least a month, and good responders were stable for a minimum of  several months.

  1. LINES 205-208: “Forty-one dogs, aged 0.6 to 13.0 years of age (median 5.8) matched the inclusion criteria. The most common breed was German Shepherd (4/41), followed by Golden Retrievers (3/41) and mixed breed dogs (3/41) (table 1). Twenty-eight dogs were male, of which 15 were intact and 13 castrated, and 14 dogs were female, of which 8 were castrated and 5 entire female dogs.”

Twenty-eight males and 14 females are 42 dogs, not 41.

Response: Thanks for noting this! We have corrected the numbers. We did orignially have 42 dogs, but one of the dogs started additional immunosuppressive treatment shortly after FMT, and was thus removed. This is probably the reason for the miscalculation

  1. LINES 223-224: “The most common histopathologic diagnoses was mild to moderate lymphocytic-plasmacytic enteritis and/or colitis in 22/35 dogs (table 2).”
    The intestinal biopsies were available from 34 dogs, not 35.

Response: Thanks for noting this! The number is corrected

  1. LINE 228: The table is difficult to read.

Response: Reviewer 1 has commented that ” Tables and figures are clear and appropriate.” We leave it to the editor to decide if the table should be edited

  1. LINES 230-244: 24 P-IRE, 4 P-IRE+ARE, 7 IRE+corticosteroids, 5 NRE, 1 FRE, 1 ARE are 42 dogs, not 41.

Response: Thanks for noting this! We have corrected the numbers. We did orignially have 42 dogs, but one of the dogs started additional immunosuppressive treatment shortly after FMT, and was thus removed. This is probably the reason for the miscalculation

  1. LINES 256-259: please summarize these results in a table: it is very confounding to read it as a unique flow of information.
    The sum of dogs is 42.

Response: A table has been added and the number corrected

  1. LINES 264-270: “The CIBDAI range (median) prior to treatment in all dogs was 2-17 (6), which decreased to 1-9 (2) after FMT.” and “Figure 2. Canine inflammatory bowel disease activity index (CIBDAI) in 41 dogs with chronic enteropathy treated with faecal microbiota transplantation (FMT) before FMT and after sequential FMT. Short horizontal lines represent median.” It is not clear what “after FMT” and “after sequential FMT” means. There are dogs with just 1 FMT, other that made many FMTs beyond the 3FMT series.

Response: We have clarified this.

  1. LINES 287-288: “Regarding histopathology reports and response to FMT, 2/4 dogs with severe inflammation of the small or large intestine had a good response, 1/4 had a short-lasting response and 1/4 had a poor response.”
    Why the consideration on response to FMT are made only on the 4 dogs with severe inflammation?

Response: We anticipated a question if the dogs with mores severe inflammation were the poor responders (which was not the case).

  1. LINE 326: “3.5 Bacterial qPCR results” This paragraph should be the 3.4, or 3.4 is missing.

Response: Thank you, we have corrected this

  1. LINE 333: “Longitudinal samples were only available from 4 dogs (Fig 5).” This sentence and the figure 5 don’t lead at any conclusion and its presence is not adding any information.

Response: We thank the reviewer for the comment. Since FMT is a new field, few data is available about follow-up microbiome responses to FMT, and whether the microbiome normalizes or not following FMT. Additionaly most of these data is based on untargeted sequencing that makes it difficult (or impossible) to compare these responses across case series.  While we have only data from 4 dogs available, we believe these descriptive longitudinal cases using a reproducible test, together with clinical long-term outcome, provide important preliminary data that can be used to better design future FMT studies.

  1. LINE 475 “The antibiotic-sparing effect of FMT in 3/3 dogs where FMT was used to avoid repeated use or reduced use of antibiotics is encouraging” and LINE 239 “Four dogs had P-IRE and antibiotic responsive enteropathy (ARE), of 239 which 3/4 dogs required frequent courses of antibiotics to control flare ups, and 1/4 dogs 240 required both corticosteroids q 24 hr, chlorambucil q 48 hr and metronidazole q 24 hr to 241 have some degree of control of clinical sign.”

The n. of animals in not coherent.

Response: We have re-phrased this section.

  1. LINES 505-506: “Our results suggest that dogs with a higher DI, indicating a more severe shift in the microbiome, may be less likely to respond favorably to FMT.”
    This sentence is not supported by results, since the dogs categories identified are misleading.

Response: There was only one dog with a short-lasting response. We have removed this dog, so the comparison is now between good and poor responders only, and  significantly different. Furthermore, we believe it is of clinical interest to compare the dogs with a very good response to FMT with the ones with a poor response, even though the number of samples is small. This descriptive data is important as baseline for future  design of prospective clinical studies/